# ExVideo: Extending Video Diffusion Models via Parameter-Efficient Post-Tuning

## Abstract

Recently, advancements in video synthesis have attracted significant attention. Video synthesis models such as AnimateDiff and Stable Video Diffusion have demonstrated the practical applicability of diffusion models in creating dynamic visual content. The emergence of SORA has further spotlighted the potential of video generation technologies. Despite advancements, the extension of video lengths remains constrained by computational resources. Most existing video synthesis models are limited to generating short video clips. In this paper, we propose a novel post-tuning methodology for video synthesis models, called ExVideo. This approach is designed to enhance the capability of current video synthesis models, allowing them to produce content over extended temporal durations while incurring lower training expenditures. In particular, we design extension strategies across common temporal model architectures respectively, including 3D convolution, temporal attention, and positional embedding. To evaluate the efficacy of our proposed post-tuning approach, we trained ExSVD, an extended model based on Stable Video Diffusion model. Our approach enhances the model's capacity to generate up to $5\times$ its original number of frames, requiring only 1.5k GPU hours of training on a dataset comprising 40k videos. Importantly, the substantial increase in video length doesn't compromise the model's innate generalization capabilities, and the model showcases its advantages in generating videos of diverse styles and resolutions. We will release the source code and the enhanced model publicly[1].

## 1 Introduction

In recent years, diffusion models (Sohl-Dickstein et al., 2015; Ho et al., 2020) have achieved outstanding results in image synthesis, significantly surpassing previous GANs (Dhariwal & Nichol, 2021). These achievements have subsequently fostered a burgeoning interest in the adaptation of diffusion models for video synthesis. Models such as Stable Video Diffusion (Blattmann et al., 2023), AnimateDiff (Guo et al., 2023), and VideoCrafter (Chen et al., 2023a) epitomize this research trajectory, showcasing the ability to produce frames that are not only coherent but also of high visual quality. These achievements underscore the practicality and potential of employing diffusion models in the field of video synthesis. With the groundbreaking results of SORA (Liu et al., 2024) reported at the beginning of 2024, the research direction of video synthesis has once again attracted widespread attention.

Although current video synthesis models are capable of producing video clips of satisfactory quality, the generated videos are generally short, and extending their duration remains a challenge. Current methodologies can be categorized into three types to generate longer videos. 1) Pre-training using long video datasets (Chen et al., 2024b; Wang et al., 2023b; Bain et al., 2021). Through extensive training with long video samples, it is foreseeable that models can improve their ability to generate longer videos. However, training with such datasets would result in prohibitively escalated costs. Consequently, given the computational constraints, current video generation models are primarily trained on short video clips. 2) Generating videos in a streaming (Kodaira et al., 2023) or sliding window (Duan et al., 2024) manner. Without further training, longer videos can be generated by stitching together several short video segments. However, this approach leads to lower video

---

[1]Project page: `https://zxqwertyuiopasdfghjk.github.io/ExVideoProjectPage/`. This page is presented anonymously, and the source code is withheld during the double-blind review process.

coherence. In addition, existing video generation models lack the capability for long-term video understanding, making the accumulation of errors inevitable. As a result, during the generation of long videos, the visual quality is prone to deterioration, manifesting as a breakdown in the imagery. 3) Frame interpolation (Huang et al., 2022; Wu et al., 2024). Video frame interpolation models offer a method to augment the frame count of generated videos. However, this approach is inadequate for extending the narrative timeframe of the video. While it increases the number of frames, maintaining the original frame rate would result in an unnatural slow-motion effect, thereby failing to extend the narrative span of the video content. These outlined challenges underscore the necessity for innovative solutions capable of overcoming the existing hurdles associated with video duration extension, without compromising video quality or coherence.

Recent breakthroughs in the development of LLMs (large language models) (Xiong et al., 2023; Xiao et al., 2023; Chen et al., 2023c) have inspired us. Notably, LLMs, despite being trained on fixed-length data, exhibit remarkable proficiency in understanding contexts of variable lengths. This flexibility is further enhanced through the integration of supplementary components and the application of lightweight training procedures, enabling the processing of exceptionally lengthy texts. Such innovations have motivated us to explore analogous methodologies within video synthesis models. In this paper, we introduce a novel post-tuning strategy, called ExVideo, specifically designed to empower existing video synthesis models to produce extended-duration videos within the constraints of limited computational resources. We have designed an extension structure for mainstream video synthesis model architectures. This framework incorporates adapter components, meticulously engineered to preserve the intrinsic generalization capabilities of the base model. Through post-tuning, we enhance the temporal modules of the model, thereby facilitating the processing of content across longer temporal spans.

In theory, ExVideo is designed to be compatible with the majority of existing video synthesis models. To empirically validate the efficacy of our post-tuning methodology, we applied it to the Stable Video Diffusion model (Blattmann et al., 2023), a popular open-source image-to-video model. Through ExVideo, we can extend the original frame synthesis capacity from a limit of 25 frames to 128 frames. Importantly, this expansion was achieved without compromising the model's distinguished generative capabilities. Additionally, the enhanced model exhibits the versatility to be seamlessly integrated with text-to-image models (Podell et al., 2023; Li et al., 2024b; Chen et al., 2023b). This synergistic amalgamation establishes robust and versatile text-to-video pipelines. This adaptability underscores the potential of our post-training technique, the source code and the extended model will be released publicly. In summary, the contributions of this paper include:

- We present ExVideo, a post-tuning technique for video synthesis models that can extend the temporal scale of existing models to facilitate the generation of long videos.

- Based on Stable Video Diffusion (SVD), we have trained an extended video synthesis model named ExSVD. This model is capable of generating coherent videos of up to 128 frames while preserving the generative capabilities of the original model.

- Through comprehensive empirical experiments, we demonstrate the feasibility of enhancing video synthesis models via post-tuning, thereby presenting an innovative approach to the training of large-scale models for extended video synthesis.

## 2 RELATED WORK

### 2.1 DIFFUSION MODELS

Diffusion models (Sohl-Dickstein et al., 2015; Ho et al., 2020) are a category of generative models that characterize the content generation as a Markov random process. Unlike GANs (Goodfellow et al., 2014), diffusion models do not require adversarial training, thus making their training process more stable. Moreover, through an iterative generation process, diffusion models are capable of producing images with exceptionally high quality. In recent years, image synthesis models based on diffusion, including Pixart (Chen et al., 2023b), Imagen (Saharia et al., 2022), Hunyuan-DiT (Li et al., 2024b), and the Stable Diffusion series (Rombach et al., 2022; Podell et al., 2023; Kang et al., 2024), have achieved impressive success. Diffusion models have given rise to a vast open-source technology ecosystem. Technologies such as LoRA (Hu et al., 2021), ControlNet (Zhang et al., 2023), DreamBooth (Ruiz et al., 2023), Textual Inversion (Gal et al., 2022), and IP-Adapter

(Ye et al., 2023) have endowed the generation process of diffusion models with a high degree of controllability, thereby meeting the needs of various application scenarios.

## 2.2 VIDEO SYNTHESIS

Given the remarkable success of diffusion models in image synthesis, video synthesis approaches based on diffusion have also been proposed in recent years. For example, by adding motion modules to the UNet model (Ronneberger et al., 2015) in Stable Diffusion (Rombach et al., 2022), AnimateDiff (Guo et al., 2023) transfers the capabilities of image synthesis to video synthesis. Stable Video Diffusion (Blattmann et al., 2023) is an image-to-video model architecture and can synthesize video clips after end-to-end video synthesis training. Unlike image synthesis models, video synthesis models require substantial computational resources since the model needs to process multiple frames simultaneously. As a result, most existing video generation models (Guo et al., 2023; Chen et al., 2023a; Wang et al., 2023a) can only produce very short video clips. For instance, AnimateDiff can generate up to 32 frames, while Stable Video Diffusion can generate a maximum of 25 frames. This limitation prompts us to explore the methodology to construct video synthesis models over longer temporal scales.

## 2.3 EXTENDING GENERATIVE MODELS

Although the existing diffusion models are trained with a fixed scale, such as Stable Diffusion being trained at a fixed resolution of $512 \times 512$, some approaches can extend them to larger scales. For instance, in image synthesis, approaches like Mixture of Diffusers (Jiménez, 2023), MultiDiffusion (Bar-Tal et al., 2023), and ScaleCrafter (He et al., 2023) can increase the resolution of generated images by altering the inference process of the UNet model in Stable Diffusion. Similar techniques have also emerged in the field of large language models. With the help of positional encoding technologies such as RoPE (Su et al., 2024) and ALiBi (Press et al., 2021), large language models can extrapolate to longer text processing tasks under the premise of training with limited-length texts. Post-tuning can further help language models achieve super-long text comprehension and generation (Xiong et al., 2023; Chen et al., 2023c). These research findings have inspired and motivate us to explore the extension of video synthesis models. We aim to endow existing video synthesis models with the capability to generate longer videos.

## 3 METHODOLOGY

In this section, we first review the architectures of mainstream video diffusion models, then discuss the methodologies we have adopted to extend the temporal modules for long video synthesis, and finally introduce the post-tuning strategy.

### 3.1 PRELIMINARIES

The huge demands of computational resources for training video synthesis models lead to a prevalent practice of adapting existing image synthesis models for video generation. This adaptation is typically achieved by incorporating temporal modules into the model for generating dynamic content. We provide a comprehensive overview of temporal module architectures as follows:

- **3D convolution** (Li et al., 2021): Convolution layers form the foundational blocks in computer vision. 2D convolution layers have been employed in the UNet (Ronneberger et al., 2015) architecture, which is widely used in diffusion models. By extending 2D convolutions into the third dimension, these layers are seamlessly adapted in video synthesis models. Research indicates that convolution layers in diffusion models exhibit a high degree of adaptability across various resolutions (Bar-Tal et al., 2023), which is a testament to their capacity for generalization.

- **Temporal attention** (Vaswani et al., 2017): In image synthesis, the importance of attention mechanisms is underscored by their contribution to the generation of images with remarkable fidelity, as evidenced by the ablation studies in latent diffusion (Rombach et al., 2022). Transferring spatial attention mechanisms to the video domain raises concerns regarding computational efficiency due to the quadratic time complexity of the attention operators.

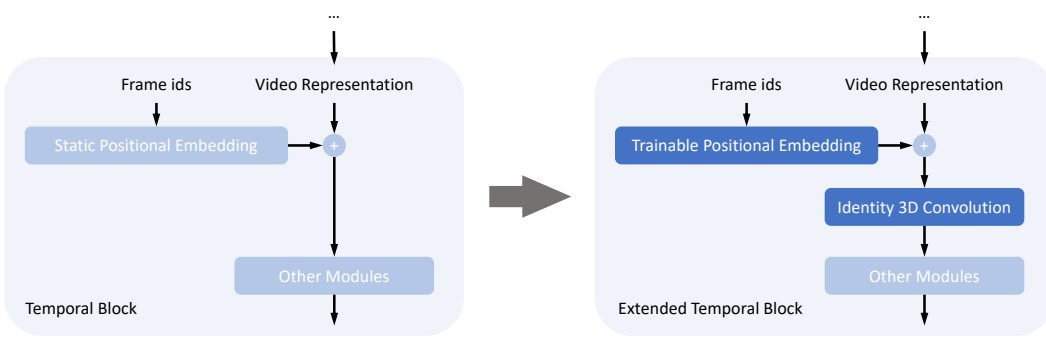

Figure 1: The architecture of extended temporal blocks in Stable Video Diffusion. We replace the static positional embedding with a trainable positional embedding and add an adaptive identity 3D convolution layer to learn long-term video features. The modifications are adaptive, preserving the original generalization abilities of the pre-trained model. All parameters outside the temporal block are fixed while training for lower memory usage.

To circumvent this computational bottleneck, advanced video synthesis models typically adopt temporal attention layers (Guo et al., 2023; Blattmann et al., 2023) that optimize efficiency by curtailing the volume of embeddings processed by each attention operator.

- **Positional embedding** (Su et al., 2024): The native attention layers cannot model the positional information in videos. Therefore, video synthesis models typically incorporate positional embeddings to enrich the embedding space with positional information. Positional embeddings can be instantiated through diverse methodologies. For example, Animate-Diff (Guo et al., 2023) opts for learnable parameters to establish positional embeddings, whereas Stable Video Diffusion (Blattmann et al., 2023) utilizes trigonometric functions to generate static positional embeddings.

### 3.2 EXTENDING TEMPORAL MODULES

Most video synthesis models are pre-trained on videos comprising only a constrained number of frames due to limited computational resources. For instance, Stable Video Diffusion Blattmann et al. (2023) is capable of generating a maximum of 25 frames, while AnimateDiff Guo et al. (2023) is limited to synthesizing image sequences of up to 32 frames. To augment these models to produce extended videos, we propose enhancements to the temporal modules within these models.

Firstly, the inherent functionality of 3D convolution layers to adaptively accommodate various scales has been previously validated through empirical studies (Jiménez, 2023; Bar-Tal et al., 2023; He et al., 2023), even without necessitating fine-tuning. Consequently, we opt to retain the 3D convolution layers in their original form to preserve these capabilities. Secondly, regarding the temporal attention modules, research on large language models has demonstrated the potential for scaling existing models to accommodate longer contextual sequences (Xiong et al., 2023; Chen et al., 2023c). Inspired by these findings, we fine-tune the parameters within the temporal attention layers during the training process to enhance their efficacy over extended frame sequences. Thirdly, for the positional embedding layers, either static or trainable embeddings cannot be directly applied to longer videos. To circumvent this pitfall while ensuring compatibility with a wide array of existing video models, we use extended trainable parameters to replace the original positional embeddings. These extended trainable positional embeddings are initialized in a cyclic pattern, drawing upon the configurations of the pre-existing embeddings. Further, drawing inspiration from various adapter models (Hu et al., 2021; Zhang et al., 2023), we incorporate an additional identity 3D convolution layer after the positional embedding layer, aimed at learning long-term information. The central unit of this 3D convolution kernel is initialized as an identity matrix, and the remaining parameters are initialized to zero. The identity 3D convolution layer ensures that, before training, there is no alteration to the video representation, thereby maintaining consistency with the original computational process.

We apply our devised extending approach to Stable Video Diffusion (Blattmann et al., 2023), which is a popular model within open-source communities for video synthesis. The comparative architectures, both pre and post-extension, are illustrated in Figure 1. Because of the fundamental similarities that underpin the construction of temporal blocks within video synthesis models, our extending approach can also be applied to various video synthesis models.

### 3.3 POST-TUNING

After extending the temporal blocks in the video synthesis models, we enhance the model's abilities to generate extended videos via post-tuning. To circumvent potential copyright concerns with video content, we employed a publicly available dataset OpenSoraPlan[2], which comprises 40,258 videos. These videos were sourced from copyright-free platforms, including Mixkit[3], Pexels[4], and Pixabay[5]. The videos in this dataset maintain a resolution of $512 \times 512$. ExVideo expands its capacity to 128 frames. Over such extended sequences, full training is deemed impractical because of the substantial computational requirements. Instead, we employed several engineering optimizations aimed at optimizing GPU memory usage. These optimizations are crucial for managing the increased computational load and facilitating efficient training within limited hardware resources:

- **Parameter freezing**: All parameters except the temporal blocks are frozen.
- **Mixed precision training** (Micikevicius et al., 2017): We deploy a mixed precision training program by converting a subset of parameters to 16-bit floating-point format.
- **Gradient checkpointing** (Feng & Huang, 2021): Gradient checkpointing is enabled in the model. By storing intermediate states during forward passes and recomputing gradients on-demand during the backward pass, this technique effectively decreases memory usage.
- **Flash Attention** (Dao, 2023): We integrate Flash Attention to enhance the computational efficiency of attention mechanisms.
- **Shard optimizer states and gradients**: We leverage DeepSpeed (Rasley et al., 2020), a library optimized for distributed training, to enable shard optimizer states and gradients across multiple GPUs.

The loss function and the noise scheduler are consistent with the original model. The learning rate is $1 \times 10^{-5}$ and the batch size on each GPU is 1. The training was conducted using only 8 NVIDIA A100 GPUs over one week. In order to ensure the stability of the training process, exponential moving averages were employed for the update of weights.

## 4 EXPERIMENTS

By integrating extended temporal modules into the original Stable Video Diffusion (SVD) model and performing post-tuning, we have developed the Extended Stable Video Diffusion (ExSVD) model. This enhanced model is capable of generating coherent videos with lengths of up to 128 frames. To validate its capabilities, we have conducted comprehensive experiments comprising three components. First, we perform a comparative analysis between our ExSVD and the original SVD model to elucidate the enhancements achieved through post-tuning. Second, we evaluate the performance of ExSVD in comparison to other publicly accessible models. Finally, we present illustrative examples to provide a tangible understanding of the model's performance.

### 4.1 EVALUATION ON EXTENDED MODEL PERFORMANCE

To assess the performance of ExVideo, we present the results of our ExSVD model and compare them with those of the original SVD model across two primary dimensions: automatic metrics and human evaluation.

---

[2]https://huggingface.co/datasets/LanguageBind/Open-Sora-Plan-v1.0.0
[3]https://mixkit.co/
[4]https://www.pexels.com/
[5]https://pixabay.com/

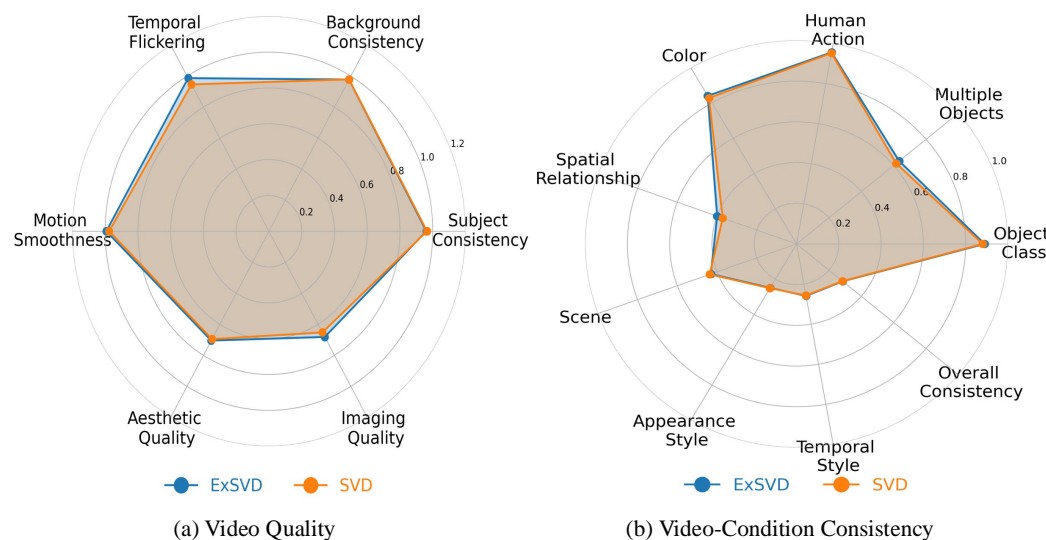

(a) Video Quality           (b) Video-Condition Consistency

Figure 2: Automatic metrics computed based on the videos generated by ExSVD and the original SVD model. When the model is extended to a larger time scale, our approach effectively preserves the original capabilities of the model.

### 4.1.1 AUTOMATIC EVALUATION

**Parameter Settings**: The comparative experiments are conducted based on VBench (Huang et al., 2024), which is a comprehensive suite of tools designed to automatically assess the quality of generated videos. Following VBench's prompt sampling methodology, we assigned 5 random seeds to each prompt for the text-to-video generation process. Initially, we employed the Hunyuan-DiT (Li et al., 2024b) text-to-image model to generate 5 images for each prompt. Subsequently, for each image, both our ExSVD and the SVD models were used to generate corresponding videos, resulting in a total of 4720 ($944 \times 5$) videos for each model. We used the DDIM (Song et al., 2020) sampler with 50 sampling steps in our experiments.

**Evaluation Metrics**: VBench assesses video performance from two broad perspectives: video quality and video-condition consistency. Video quality focuses on the perceptual quality of the synthesized video, including temporal quality and frame-wise quality. This category is composed of seven sub-metrics: subject consistency, background consistency, temporal flickering, motion smoothness, dynamic degree, aesthetic quality, and imaging quality. Video-condition consistency focuses on whether the synthesized video aligns with the user-provided guiding condition (text prompt) and includes metrics from semantic and style dimensions. This category is composed of nine sub-metrics: object class, multiple objects, human action, color, spatial relationship, scene, appearance style, temporal style, and overall consistency. It is important to note that the dynamic degrees in the original implementation of VBench are sensitive to frames per second (FPS) and the total number of frames. Our findings indicate that this metric can significantly impact the overall score, potentially resulting in biased comparisons between different models. Our model will achieve an exceptionally high score when we use a high FPS, which is ultimately inconsequential. We will address this issue in future research, as there currently are no valid metrics available for evaluation.

**Quantitive Results**: Based on the generated videos from our ExSVD and the original SVD model, we have calculated all the automatic metrics in VBench. The results are depicted in Figure 2. From the perspective of video quality, most metrics of the ExSVD model are on par with those of the SVD model, indicating that ExSVD does not degrade the video quality of the original SVD model. In the temporal flickering dimension, the ExSVD model demonstrates superior performance. This enhancement is primarily attributed to the extended temporal block in the ExSVD model, which bolsters the model's motion prediction capabilities. After post-tuning, the ExSVD model exhibits enhanced temporal consistency, resulting in fewer flickering phenomena and thus a higher temporal flickering score. From the perspective of video-condition consistency, the performances of the

Table 1: User preference between SVD and ExSVD.

| ExSVD is better | Tie | SVD is better |
|:---:|:---:|:---:|
| **48.35%** | 16.88% | 34.77% |

Table 2: Comparison with other open-accessible models on VBench.

| Model | Total Score | Subject Consistency | Background Consistency | Temporal Flickering | Motion Smoothness | Aesthetic Quality | Imaging Quality | Object Class |
|---|---|---|---|---|---|---|---|---|
| OpenSora | 79.23% | 94.45% | 97.90% | 99.47% | 98.20% | 56.18% | 60.94% | 83.37% |
| AnimateDiff | 80.27% | 95.30% | 97.68% | 98.75% | 97.76% | 67.16% | 70.10% | 90.90% |
| VideoCrafter2 | 80.44% | 96.85% | 98.22% | 98.41% | 97.73% | 63.13% | 67.22% | 92.55% |
| Pika | 80.69% | 96.94% | 97.36% | 99.74% | 99.50% | 62.04% | 61.87% | 88.72% |
| T2V-Turbo | 81.01% | 96.28% | 97.02% | 97.48% | 97.34% | 63.04% | 72.49% | 93.96% |
| CogVideoX | 81.61% | 96.23% | 96.52% | 98.66% | 96.92% | 61.98% | 62.90% | 85.23% |
| LaVie | 81.75% | 97.90% | 98.45% | 98.76% | 98.42% | 67.62% | 70.39% | 97.52% |
| Kling | 81.85% | 98.33% | 97.60% | 99.30% | 99.40% | 61.21% | 65.62% | 87.24% |
| SVD | 81.73% | 96.52% | 97.89% | 94.66% | 97.58% | 69.56% | 65.25% | 88.10% |
| ExSVD (ours) | **81.91%** | 96.11% | 97.79% | 98.71% | 99.31% | 70.47% | 68.16% | 88.99% |
| Model | Multiple Objects | Human Action | Color | Spatial Relationship | Scene | Appearance Style | Temporal Style | Overall Consistency |
| OpenSora | 58.41% | 85.80% | 87.49% | 67.51% | 42.47% | 23.89% | 24.55% | 27.07% |
| AnimateDiff | 36.88% | 92.60% | 87.47% | 34.60% | 50.19% | 22.42% | 26.03% | 27.04% |
| VideoCrafter2 | 40.66% | 95.00% | 92.92% | 35.86% | 55.29% | 25.13% | 25.84% | 28.23% |
| Pika | 43.08% | 86.20% | 90.57% | 61.03% | 49.83% | 22.26% | 24.22% | 25.94% |
| T2V-Turbo | 54.65% | 95.20% | 89.90% | 38.67% | 55.58% | 24.42% | 25.51% | 28.16% |
| CogVideoX | 62.11% | 99.40% | 82.81% | 66.35% | 53.20% | 24.91% | 25.38% | 27.59% |
| LaVie | 64.88% | 96.40% | 91.65% | 38.68% | 49.59% | 25.09% | 25.24% | 27.39% |
| Kling | 68.05% | 93.40% | 89.90% | 73.03% | 50.86% | 19.62% | 24.17% | 26.42% |
| SVD | 61.55% | 95.20% | 82.81% | 37.17% | 43.68% | 25.12% | 25.71% | 28.39% |
| ExSVD (ours) | 63.38% | 95.60% | 84.04% | 39.94% | 43.14% | 24.91% | 25.96% | 28.55% |

ExSVD and SVD models are generally consistent. In the spatial relationship and multiple objects metrics, ExSVD achieves higher scores. This indicates that ExSVD is capable of fully leveraging the text-to-image model to synthesize realistic videos based on the generated images. Overall, ExVideo enhances the total number of frames and duration of videos without compromising video quality.

**Human Evaluation**: In addition to the automatic metrics evaluation, we conducted a human preference experiment involving 30 participants to facilitate a comparative analysis between the SVD and our ExSVD model. In each evaluation session, we randomly selected two videos that both corresponded to the same prompt and presented them to the participants. Participants were instructed to choose from one of three options: "Left is better", "Tie", or "Right is better", without disclosing the names of the models. Each participant evaluated up to 30 randomly selected video pairs. The results, detailed in Table 1, indicate that our ExSVD model outperformed the SVD model in terms of human preference, achieving a win rate of 48.35% compared to SVD's win rate of 34.77%.

### 4.2 COMPARISON WITH PUBLICLY ACCESSIBLE MODELS

We further evaluated the performance of ExSVD in comparison to other publicly available models. To facilitate a comprehensive analysis that incorporates both text-to-video and image-to-video models, we designed a text-to-video pipeline that integrates ExSVD with Hunyuan-DiT. This allows for a uniform assessment of the models across the text-to-video task. The parameter settings of this evaluation were consistent with those outlined in the previous subsection.

**Baseline Models**: For a thorough evaluation of our models, we selected top-performing video synthesis models for comparison. The chosen models include OpenSora (Zheng et al., 2024), Animate-Diff (Guo et al., 2023), VideoCrafter2 (Chen et al., 2024a), Pika (Pika, 2024), T2V-Turbo (Li et al., 2024a), CogVideoX (Yang et al., 2024), LaVie (Wang et al., 2023a), and Kling (Team, 2024).

**Quantitive Results**: The results are summarized in Table 2, where the metrics of the baseline models are collected from the original VBench leaderboard. Due to the variability of the dynamic degree metric with respect to frame and FPS, we have opted not to include it in the table. In comparison to other models, our ExSVD outperforms the competition, achieving the highest overall score and ex-

celling in the aesthetic quality, temporal style, and overall consistency metrics. Additionally, ExSVD exhibits competitive performance in the dimensions of temporal flickering, motion smoothness, and multiple objects.

### 4.3 CASE STUDY

#### 4.3.1 VISUAL COMPARISON

We present a series of video examples generated by ExSVD and other video synthesis models to facilitate an intuitive comparison of their performance. The illustrative results from these models are displayed in Figure 3. We highly recommend readers watch the videos on our anonymous project page. A noteworthy observation from this comparison is that the majority of existing video synthesis models tend to produce videos characterized by limited motion dynamics. In contrast, our extended model, which benefits from post-tuning processes applied over extended temporal durations, exhibits a markedly improved ability to generate videos with significant movement. In the second example, Kling, a competitive close-source model, demonstrates the capability to generate realistic videos, but it is unable to generate the astronaut in the style of Van Gogh. This disparity in performance highlights the advanced generative capabilities of our model.

#### 4.3.2 GENERALIZATION ABILITIES

Although the model is trained at a fixed resolution and exclusively utilizes realistic video datasets, the extended variant demonstrates remarkable capabilities, allowing for the generation of videos across a spectrum of resolutions and styles. To rigorously assess the performance of ExSVD, we conduct additional evaluations across various resolutions. Figure 4 illustrates several generated video examples that underscore the model's capacity to generate videos in various resolutions and aspect ratios. This adaptability ensures that the generated videos maintain visual integrity regardless of the resolution parameters. Furthermore, Figure 5 presents an array of stylistic variations, further emphasizing the model's versatility in accommodating diverse artistic expressions. These examples underscore the robustness and generalizability of ExSVD, offering flexibility in video generation across varying contexts.

## 5 LIMITATIONS

While ExVideo can enhance the capabilities of video diffusion models, the post-tuned version continues to be constrained by the inherent limitations of its foundational model. Notably, the extended Stable Video Diffusion struggles to accurately synthesize human portraits, leading to frequent instances of truncated frames. To develop a model capable of synthesizing high-quality long videos, it is imperative to train a robust base model. Nevertheless, due to limitations in resources, we are unable to independently pre-train a large video synthesis model. Consequently, we eagerly anticipate the release of open-source models in the future to advance our research endeavors.

## 6 CONCLUSIONS AND FUTURE WORK

In this paper, we delve into the enhancement of video diffusion models through post-tuning. Specifically, we propose a post-tuning approach called ExVideo, which can extend the duration of generated videos and release the potential of video synthesis models. Based on Stable Video Diffusion, our approach achieves a quintupling in the number of frames, while preserving the original generalization abilities. ExVideo is designed within the constraints of limited computational resources, thus it is exceptionally memory-efficient. By integrating this method with other open-source technologies, we facilitate pipelines conducive to the production of high-quality videos. However, despite the advancements achieved through post-tuning, the enhanced model remains inherently constrained by the limitations of the base model. Looking ahead, we are committed to furthering our exploration of video synthesis models through post-tuning methodologies. This will include the application of ExVideo across a broader range of model architectures, as well as the training of these models on larger and more varied datasets.

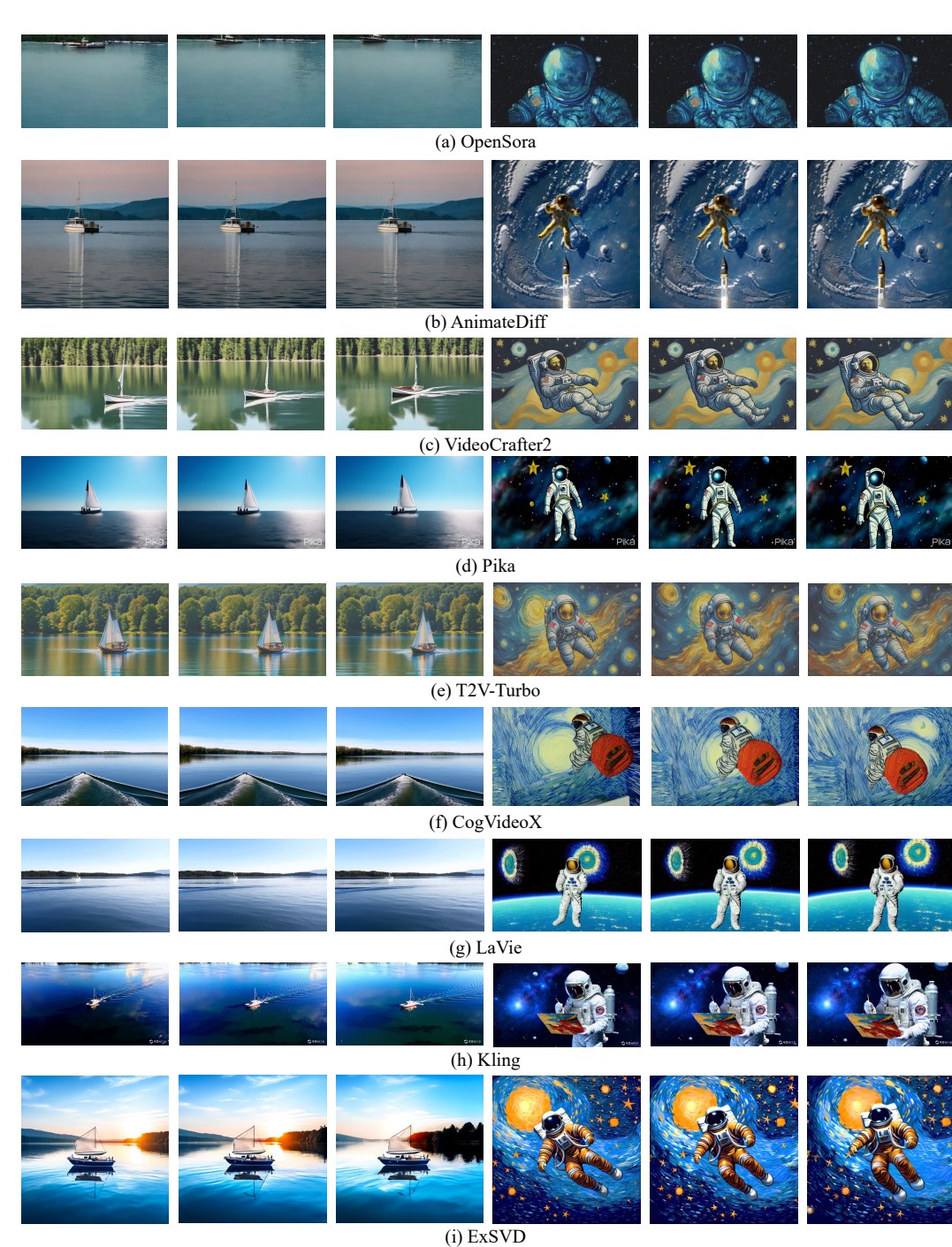

(a) OpenSora

(b) AnimateDiff

(c) VideoCrafter2

(d) Pika

(e) T2V-Turbo

(f) CogVideoX

(g) LaVie

(h) Kling

(i) ExSVD

Figure 3: Visual comparisons of text-to-video results from several existing video synthesis models and our Extended model. The prompts are "a boat sailing smoothly on a calm lake" and "an astronaut flying in space, Van Gogh style". In our pipeline, the first frame is generated by Hunyuan-DiT, and ExSVD generates the video according to the first frame. We highly recommend readers watch the videos on our anonymous project page: `https://zxqwertyuiopasdfghjk.github.io/ExVideoProjectPage/`.

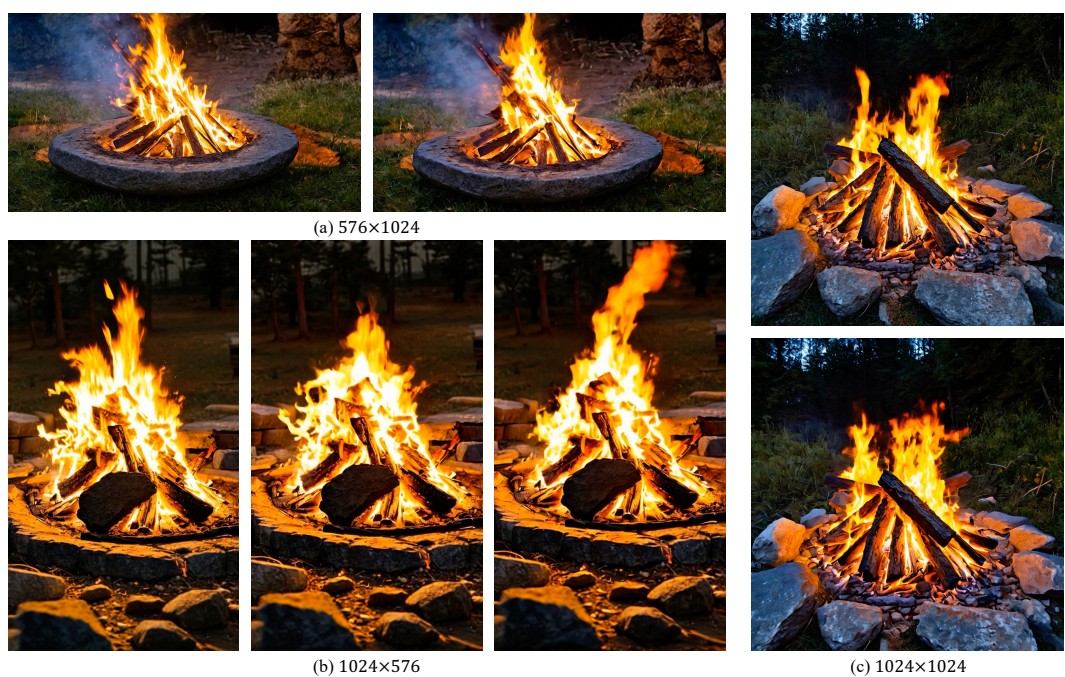

(a) 576×1024

(b) 1024×576

(c) 1024×1024

Figure 4: Video examples in various resolutions. The first frame is generated by Stable Diffusion 3, and the prompt is "bonfire, on the stone".

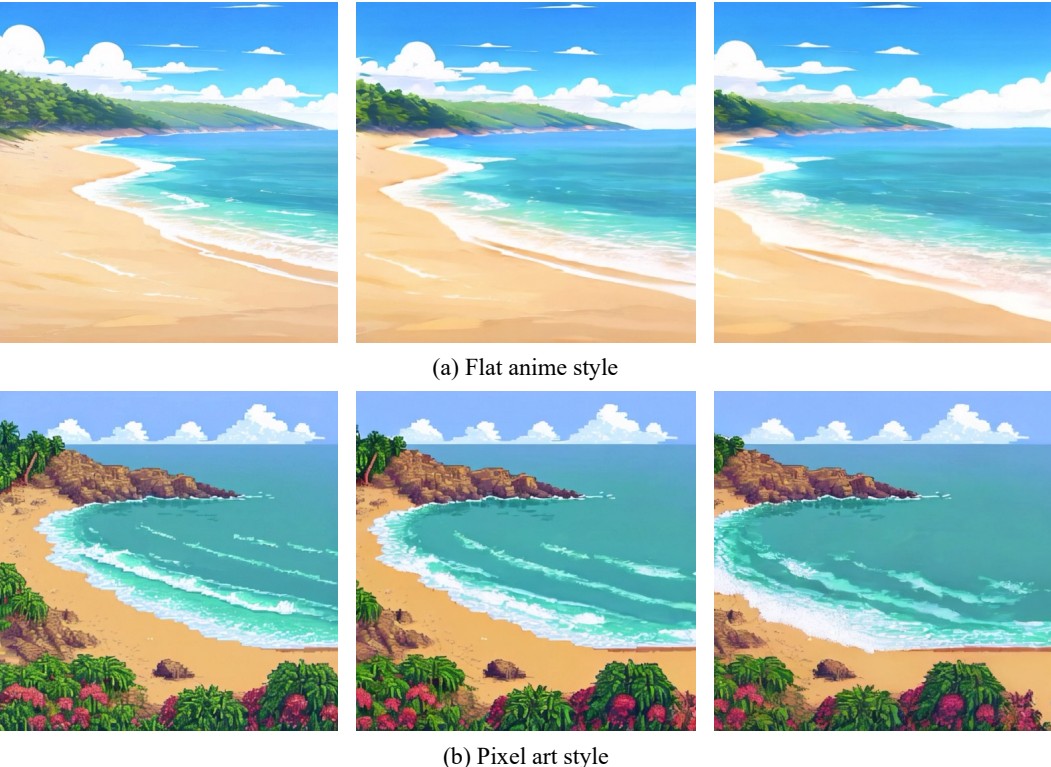

(a) Flat anime style

(b) Pixel art style

Figure 5: Examples in various styles generated by ExSVD, where the first frame is generated by Stale Diffusion 3. The prompt is "A beautiful coastal beach in spring, waves lapping on sand", followed by the description of style.

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

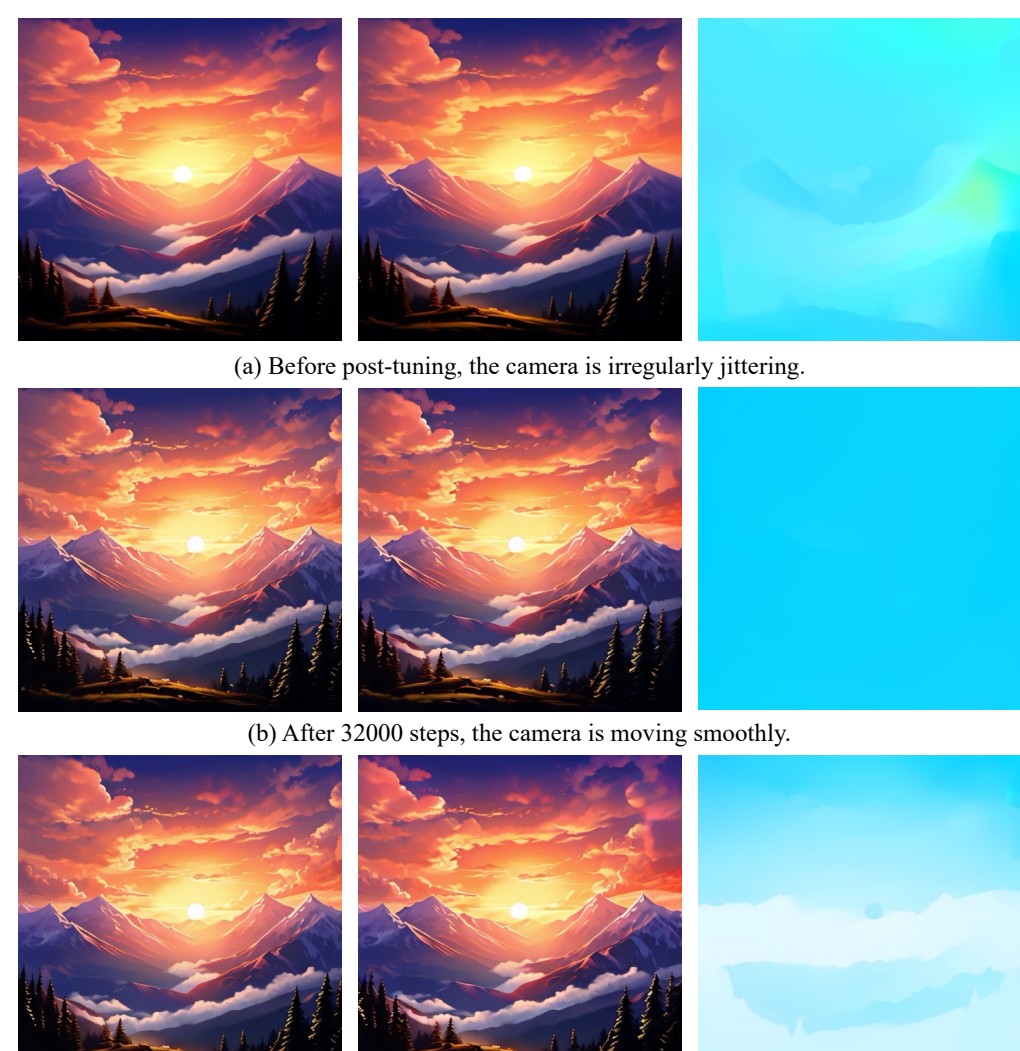

(a) Before post-tuning, the camera is irregularly jittering.

(b) After 32000 steps, the camera is moving smoothly.

(c) After 64000 steps, complex motion emerges.

Figure 6: Video examples in different training phases. The first frame is generated by Hunyuan DiT, and the prompt is "sunset, mountains, clouds". We present the optical flow to visualize the motion, where pixels with similar colors are moving in similar directions.

## A  APPENDIX

### A.1  VISUALIZATION OF TRAINING PROCESS

We investigated the evolution of the model's capabilities during the training process. Figure 6 presents the generated videos that exemplify the model's performance at three distinct phases of training. It is difficult to present the dynamics using still images, thus we present the optical flow, computed by RAFT (Teed & Deng, 2020), to the right of each example for a clearer demonstration of motion. Initially, before training, the extended model architecture was solely capable of guaranteeing the structural integrity of the video frames, which suffered from pronounced jittering artifacts. Progressing through the training, after 32,000 steps, the model began to produce videos displaying smooth camera movements. With continued training up to 64,000 steps, the model further advanced to create complex motions, such as clouds and mountains moving with nuanced, layered speed. The model effectively understands the depth and spatial relationships within the scene. This example intuitively illustrates the process of the model learning long-term information.

