# OpenReview forum: "ExVideo: Extending Video Diffusion Models via Parameter-Efficient Post-Tuning"
_ICLR.cc/2025/Conference — Submitted to ICLR 2025_

### Official Review · Reviewer_Lib6 · 2024-10-31

**Soundness:** 2
**Presentation:** 2
**Contribution:** 1
**Rating:** 3
**Confidence:** 4

**Summary:**

This paper presents ExVideo, a post-tuning method for video synthesis models, to extend the video length in generation with limited resources. The authors extend the temporal modules of SVD and adopt efficient tuning methods for training at limited costs,

**Strengths:**

1. The generated videos seem to have a good quality.
2. Easy to use, applicable to existing models.

**Weaknesses:**

1. Limited Contribution

The contribution of the proposed method is limited; the novel part of this work is two components, using additional trainable parameters for positional embeddings, and using an additional 3D convolutional layer with a tailored initialization. Simply adding learnable parameters cannot be a contribution, without theoretical / experimental justification of its necessity. Specifically, I cannot understand why the additional (identity) 3D convolutional layer is necessary, which I believe the merit of using it would simply come from additional parameters. Likewise, I think adding trainable positional embeddings could be further explained on its necessity and efficacy, with comparison to existing methods.

In the intro, the authors address the difficulty of tuning video synthesis models for extended length generation, in terms of computational constraints. However, their proposed (temporal) modules are irrelevant to this matter, but rather introduces additional parameters, which would increase the overall cost. What makes the proposed method efficient is, to my understanding, the methods used for post-tuning, described in Section 3.3. Yet, the techniques used here are from prior research, and not a novel contribution of this paper. If the authors were to argue that their method is efficient, I believe there should have been a novel design for computational savings, which cannot be found in the current paper.


2. Experimental results

- The experiments do not reflect the main argument of this paper. In Fig.3, the generation results with comparison to existing T2V methods are presented. However, I believe this does not show how well ExVideo successfully extends the generation length of videos. The baseline methods all show good temporal coherency and excellent visual quality. The most noticeable difference between the baselines and the proposed model is text-alignment, which is not the main argument of the paper. The better text-alignment may be possibly coming from a stronger baseline, SVD. To highlight the effectiveness of the proposed method, experiments on generation quality when extending the video length with comparison to SVD would have been necessary.
In addition, in order to claim the main argument of the paper, better extension ability of video sequence, more experiments on how existing methods fail when the generation length gets longer should have been presented. (e.g., Generation quality of latter timestep frames collapsing) However, according to the qualitative results in Fig.3, existing models do not seem to be struggling with video generation of extended length, and the only difference I can find with comparison to baseline methods is text-alignment, which is presumed to be due to use of advanced baseline.

- Quantitatively, the results do not seem to be strong enough. Using 8 A100 GPUs over a week of training does not seem to be cost-efficient, but considering the cost, the improvement seems to be marginal.

- Lack of ablation studies. The paper currently does not discuss the ablation study of the proposed components, which makes it hard to verify the effectiveness of the method. For instance, the authors could have done an ablation study on how the initialization scheme in identity 3D convolutional layer affects the training, or on how switching the positional embeddings affect the performance.

**Questions:**

Please see the Weaknesses above.

---

### Official Review · Reviewer_YubY · 2024-11-02

**Soundness:** 1
**Presentation:** 2
**Contribution:** 1
**Rating:** 3
**Confidence:** 4

**Summary:**

This paper propose ExVideo, a post-tuning methodology for video synthesis models. Particularly, it designs extension strategies across common temporal model architectures respectively, including 3D convolution, temporal attention, and positional embedding. Experimental results demonstrates the effectiveness of proposed method.

**Strengths:**

The paper writing is clear and easy to understand. The visual results are pleasing.

**Weaknesses:**

1. Lack of novelty. This paper proposes extending the pre-trained SVD to generate more frames by adding an additional 3D convolution and a trainable positional embedding. These techniques are commonly used in modern video generation model architectures. Additionally, Section 3.3 includes practical training settings and parameters with no algorithm-level contribution, which should be moved to the appendix.
2. Unclear Motivation and Insufficient Comparison. The motivation of this paper is not clear, and it lacks corresponding comparisons. The authors mention that the original SVD can only generate 25 frames, and this paper proposes extending the generated videos to 128 frames. However, what is the problem with directly fine-tuning the original model on 128 frames? Is the proposed additional 3D convolution necessary? Does it save much memory or make the training procedure converge faster? If so, please provide corresponding numerical comparison results.
3. Lack of Implementation Details. Since the core contributions of this paper are the proposed "Identity 3D Convolution" and "Trainable Positional Embedding," please illustrate and describe the implementation details of these components, preferably with formulas and detailed illustrations.
4. Paper Layout Issues. Figures 3 and 4 are placed at the end of the paper, which is not visually pleasing.

**Questions:**

My main concern is the unclear motivation and insufficient comparison, please refer it at point 2 in Weaknesses.

---

### Official Review · Reviewer_SfLr · 2024-11-02

**Soundness:** 2
**Presentation:** 1
**Contribution:** 2
**Rating:** 3
**Confidence:** 4

**Summary:**

The paper titled "ExVideo: Extending Video Diffusion Models via Parameter-Efficient Post-Tuning" introduces a novel post-tuning methodology aimed at enhancing the capabilities of current video synthesis models to generate longer videos with lower training costs. The authors propose ExVideo, which is designed to extend the temporal scale of existing video synthesis models, allowing them to produce content over extended durations without compromising video quality or coherence. The paper demonstrates the efficacy of ExVideo by training an extended model, ExSVD, based on the Stable Video Diffusion model. The authors claim that ExSVD can generate up to 5 times its original number of frames with only 1.5k GPU hours of training on a dataset of 40k videos. The paper also discusses the compatibility of ExVideo with various temporal model architectures, including 3D convolution, temporal attention, and positional embedding.

**Strengths:**

Originality: The paper presents an effective approach to extend the capabilities of video synthesis models to generate longer videos, which is an original contribution in the field of video generation.

Quality: The methodology appears to be well-thought-out, with a clear description of the technical details and the design of the extension strategies for different temporal model architectures. The paper also provides a comprehensive evaluation of the proposed method, including both automatic metrics and human evaluations.

Clarity: The paper is well-organized and clearly written. The introduction sets the stage for the problem, the methodology section explains the approach in detail, and the experiments section provides a clear demonstration of the effectiveness of ExVideo. The figures and tables are well-designed and aid in understanding the content.

Significance: The work is relatively significant as it addresses a critical challenge in video synthesis, which is the ability to generate longer, high-quality videos. The proposed solution has the potential to impact various applications, including content creation, entertainment, and beyond.

**Weaknesses:**

Generalization to Other Models: While the paper claims that ExVideo is designed to be compatible with the majority of existing video synthesis models, the evaluation is primarily based on the Stable Video Diffusion model. More evidence is needed to support the generalization of ExVideo to other models.

Comparison with Other Long Video Generation Models: The paper does not introduce or compare with other existing models that focus on long video generation, particularly in terms of frame consistency over extended durations. Such comparisons are crucial to establish the relative advantages and limitations of ExVideo, especially concerning the maintenance of coherence and quality in long videos.

Lack of Ablation Study: An ablation study is missing in the paper, which would be valuable to understand the contribution of each component of the ExVideo methodology. Such a study could provide insights into the effectiveness of the proposed extensions to the temporal modules and the impact of post-tuning on the overall performance.

Novelty of the Methodology: The paper positions ExVideo as an innovative approach to extend video generation models. However, the methodology described does not appear to introduce any novel techniques but rather builds upon existing model architectures. The paper could benefit from a more detailed discussion of how ExVideo differs from and improves upon current state-of-the-art methods.

Clarity on Training and Inference Algorithm: The paper lacks a clear and detailed explanation of the exact algorithms used for training and inference of longer videos. The process by which the extended temporal modules are integrated and the specifics of the post-tuning process are not explicitly outlined. The authors should provide a more detailed description of these processes to ensure that the paper is accessible to readers who may wish to build upon this work.

**Questions:**

Scalability: How does ExVideo perform as the model scales up in terms of complexity and the number of parameters? Are there any limitations to the model's scalability?

Model Limitations: The paper mentions that the extended Stable Video Diffusion struggles with accurately synthesizing human portraits. Could the authors elaborate on the nature of these issues and any potential solutions or workarounds?

Dataset Diversity: Are there plans to train ExVideo on other datasets to evaluate its robustness across different video content?

---

### Official Review · Reviewer_TLo4 · 2024-11-04

**Soundness:** 3
**Presentation:** 3
**Contribution:** 3
**Rating:** 5
**Confidence:** 4

**Summary:**

This paper introduces a post-tuning method to enhance current video generation models, allowing them to produce longer videos with lower training costs. It leverages 3D convolution layers and temporal attention, enabling the video generation up to 5 times longer than the original length without compromising quality.

**Strengths:**

1. The writing is excellent—clean, clear, and concise.

2. Most current video generation methods demand substantial computational resources for training. However, this post-tuning method requires significantly fewer resources, such as 8 A100 GPUs for just one week.

**Weaknesses:**

1. My major concern is the novelty. The proposed methods extends the original SVD by adding an identity 3D convolution layers in the temporal block. The technique novelty is limited.

2. The experiments are not comprehensive.
(a) For the user study. The authors only compared with SVD. Authors should compare with more methods.
(b) I am a little confused of the user study setting. SVD generate videos with 25 frames, while ExSVD generates videos with 128 frames. Users can easily find the video generated by the new method. Is it a fair comparison?
(c) I have the same concerns of the results on VBench. How many frames does the ExSVD generate? Other methods, e.g. SVD, Kling, generate short videos. Are these evaluation metrics also sentensive to the number of frames?


Minor.
1. Table 2 only labels the best result in the total score column as bolden. The authors should label more results.

**Questions:**

Please refer to the weakness section.

---

### Meta-Review · Area_Chair_iZZ9 · 2024-12-21

**Metareview:**

Reviewers unanimously recommend rejection of the paper, citing lack of novelty, clarity in presentation, lack of comparisons, among other things.

Authors did not participate in discussion phase, and as such, the initial scores persist.

There are a handful of good bits of feedback for improving the current manuscript in the provided reviews, we encourage the authors to revise their manuscript and re-submit.

**Additional Comments On Reviewer Discussion:**

Reviews began negative and remained negative due to a lack of rebuttal.

---

### Decision · Program_Chairs · 2025-01-22

Reject